# Dispersal of PRC1 condensates disrupts polycomb chromatin domains and loops

Iain Williamson, Shelagh Boyle, Graeme R Grimes, Elias T Friman, Wendy A Bickmore

Polycomb repressive complex 1 (PRC1) strongly influences 3D genome organization, mediating local chromatin compaction and clustering of target loci. Several PRC1 subunits have the capacity to form biomolecular condensates through liquid–liquid phase separation in vitro and when tagged and over-expressed in cells. Here, we use 1,6-hexanediol, which can disrupt liquid-like condensates, to examine the role of endogenous PRC1 biomolecular condensates on local and chromosome-wide clustering of PRC1-bound loci. Using imaging and chromatin immunoprecipitation, we show that PRC1-mediated chromatin compaction and clustering of targeted genomic loci—at different length scales—can be reversibly disrupted by the addition and subsequent removal of 1,6-hexanediol to mouse embryonic stem cells. Decompaction and dispersal of polycomb domains and clusters cannot be solely attributable to reduced PRC1 occupancy detected by chromatin immunoprecipitation following 1,6-hexanediol treatment as the addition of 2,5-hexanediol has similar effects on binding despite this alcohol not perturbing PRC1-mediated 3D clustering, at least at the sub-megabase and megabase scales. These results suggest that weak hydrophobic interactions between PRC1 molecules may have a role in polycomb-mediated genome organization.

## Introduction

The spatial organisation of biochemical reactions in the cell nucleus is fundamental to genome regulation. The concentration of nucleic acids and proteins in membraneless compartments, or condensates, can increase the kinetics of, and modulate the outcomes of, biochemical reactions (Holehouse & Pappu, 2018). Condensates have been implicated in the regulation of a multitude of nuclear functions including DNA repair, replication, transcription, RNA processing, and epigenetic regulation (Banani et al, 2017). Many biophysical mechanisms can result in the formation of condensates, but one receiving a lot of attention currently is liquid–liquid phase separation (LLPS). Proteins that tend to undergo LLPS often have regions that participate in multivalent weak interactions that are intrinsically disordered, or that favour oligomerisation (Banani et al, 2017).

Chromatin can act as a scaffold for phase separation, and chromatin-binding proteins have been implicated in the formation of phase-separated chromatin compartments (Bajpai et al, 2021). Interest in this area was propelled by observations of the LLPS properties of the chromobox (Cbx) containing heterochromatin protein 1α (HP1α)/Cbx5 (Larson et al, 2017; Strom et al, 2017). HP1 proteins recognise and bind to histone H3 tri-methylated at lysine 9 (H3K9me3) through their Cbx domain. However, the role of LLPS in the compaction of H3K9me3-marked constitutive heterochromatin has been questioned and instead it has been suggested that the ability of HP1α to bridge between H3K9me3-modified nucleosomes is sufficient to drive the compaction of constitutive heterochromatin (Erdel et al, 2020).

Cbx domain–containing proteins are also important in the formation of facultative heterochromatin. Cbx2,4,6,7, and 8 are components of the canonical polycomb repressive complex 1 (cPRC1) and are responsible for the targeting of this complex to H3K27me3-modified chromatin, deposited by the PRC2 complex. Cbx2 has been shown to be able to phase separate in vitro (Plys et al, 2019) and when ectopically expressed in mammalian cells (Tatavosian et al, 2019), although its behaviour when induced to oligomerise in cells is more complex (Eeftens et al, 2021). In addition, other components of PRC1 have the ability to form condensates under those conditions and indeed the polyhomeotic components of cPRC1 are of particular interest because their sterile alpha motif domain, known to drive oligomerisation (Isono et al, 2013), can phase separate in vitro, form condensates in cells (Seif et al, 2020; Eeftens et al, 2021), and can compact chromatin (Kundu et al, 2017). Most of these studies have analysed the properties of cPRC1 components in vitro, or when tagged and over-expressed in cells. There has been little detailed analysis of the role of weak multivalent interactions in modulating the local and long-range chromatin conformation at target loci of the endogenous cPRC1 complex.

Both imaging and chromosome conformation capture analyses have shown that in mouse embryonic stem cells (mESCs), polycomb acts not only to locally compact target loci (Eskeland et al, 2010; Kundu et al, 2017; Boyle et al, 2020) but also to bring together distant target loci (Isono et al, 2013; Schoenfelder et al, 2015; Vieux-Rochas

MRC Human Genetics Unit, Institute of Genetics and Cancer, University of Edinburgh, Edinburgh, UK

Correspondence: Iain.Williamson@ed.ac.uk; Wendy.Bickmore@ed.ac.uk

et al, 2015; Kundu et al, 2017; Boyle et al, 2020). This clustering brings multiple cPRC1 target loci close together in nuclear space and occurs over genomic distances of 10–100 s of Mb—far beyond the scale of topologically associating domains (TADs) (Boyle et al, 2020). The same is not seen for targets of variant PRC1 complexes and that lack Cbx components (Boyle et al, 2020). Consistent with this, long-range polycomb associations do not depend on cohesin (Rhodes et al, 2020).

Here, we investigate the role of multivalent weak interactions in the ability of endogenous polycomb complexes to direct 3D chromatin organisation in mESCs. Using DNA fluorescence in situ hybridisation (DNA-FISH), we assay chromatin compaction of single large (>100 kb) polycomb-bound domains and short- (100 kb – 2 Mb) and long-range (10s of megabases) clustering of polycomb target loci in cells treated with two aliphatic alcohols, one with and one without the capacity to disrupt biomolecular condensates, and in cells allowed to recover after alcohol treatment. We show that the perturbation of cPRC1 condensates reversibly disrupts chromatin compaction and clustering of polycomb target loci. Using chromatin immunoprecipitation followed by deep sequencing (ChIP-seq), we show that these effects cannot be solely explained by the loss of cPRC1 binding at target genomic loci. We suggest that multivalent weak interactions between polycomb complexes drive the ability of cPRC1 to direct 3D genome organisation.

# Results

## Optimizing hexanediol treatment of mESCs

A significant factor hindering the investigation of phase separation in cells is the limited tools available to modulate the molecular interactions between endogenous macromolecules that drive these transitions. Many biomolecular condensates and nuclear bodies are sensitive to the aliphatic alcohol 1,6-hexanediol (1,6-HD), which can disrupt weak multivalent hydrophobic interactions, and 1,6-HD has been used to discriminate liquid-like condensates from those with more solid- or gel-like properties (Kroschwald et al, 2017). The aliphatic alcohol 2,5-hexanediol (2,5-HD) does not show this property (Lin et al, 2016; Kroschwald et al, 2017).

The addition of 1,6-HD to several human and mouse cell lines at concentrations ranging from 1.5% to over 10%, and with treatments lasting from less than a minute to 30 min, has been used to disrupt nuclear biomolecular condensates, including those implicated in chromatin states. However, some results have been conflicting and indicative of irreversible changes in chromatin and nuclear organisation (Itoh et al, 2021; Liu et al, 2021; Ulianov et al, 2021). We therefore tested a range of conditions for treating mESCs with 1,6-HD (2%, 5%, and 10% concentrations for 5, 10, and 15 min durations) before fixation to minimise harmful side effects and prevent excessive cell death. For mESCs growing in tissue-culture flasks and on microscope glass slides, we found that both 5% and 10% concentrations of 1,6-HD caused excessive cell death. Visually, cells grown in flasks treated with 2% 1,6-HD or 2,5-HD appeared normal up to 10 min treatment duration with minimal cell death, but for cells growing on glass slides optimal treatment duration was

reduced to 5 min (Figs 1A and S1A). The morphology of nuclei in mESCs treated with 2% 1.6-HD or 2,5-HD for 5 min appeared to be no different to untreated cells and cells allowed to recover for over 1 h post-treatment, with chromocentres readily apparent (Fig 1B), and nuclear size was also unaffected by HD treatment (Figs 1C and S1B and Table S1). Therefore, for all subsequent experiments, we treated cells with 2% 1,6-HD or 2,5-HD for 5 min, which corresponds well with the optimal parameters for 1,6-HD treatment of mESCs reported in a recent study (Liu et al, 2021).

## 1,6-HD but not 2,5-HD treatment reversibly decompacts Hox clusters

Components of the polycomb repressive complex 1 (PRC1) have been shown to be capable of phase separating in vitro, or when ectopically expressed (Plys et al, 2019; Tatavosian et al, 2019; Seif et al, 2020). We therefore sought to determine whether the chromatin changes induced by cPRC1 binding at endogenous PRC1 target loci in mESCs are sensitive to disruption by 1,6-HD.

The four paralogous Hox loci, containing densely packed arrays of homeobox genes, are the largest polycomb targets in mESCs, forming compact chromatin domains of over 100 kb in size that are dependent on PRC1 and PRC2 (Eskeland et al, 2010; Williamson et al, 2014; Kundu et al, 2017; Boyle et al, 2020). As an example, Hi-C analysis identifies highly enriched interactions across the HoxD region in mESCs, including the neighbouring *Evx2* gene, that correspond with the extent of H3K27me3 and RING1B (PRC1 subunit) occupancy (Fig 2A, upper heatmap and ChIP-seq tracks). This enriched interaction domain is lost in mESCs without functional PRC1 (*Ring1B$^{-/-}$*) (Fig 2A, lower heatmap).

We have previously been shown by 3D FISH that loss of PRC2 or PRC1 results in a significant visible increase in spatial distance between the 3′ and 5′ ends of the HoxD cluster in nuclei, indicative of localised chromatin decompaction, whereas the non-polycomb target region upstream of HoxD (limb enhancers GCR to *Lnp*) is unaffected (Eskeland et al, 2010; Williamson et al, 2014). To determine whether polycomb-mediated chromatin compaction is dependent on weak multivalent interactions, we measured the interprobe distances across HoxD (probes Evx2 and Hoxd3) and the control region (between GCR and Lnp) (Fig 2A, B, and D) in cells treated with 1,6-HD and 2,5-HD, compared with untreated cells and cells allowed to recover in normal media for at least 1 h post 1,6-HD treatment. Interprobe distances between the 3′ and 5′ ends of HoxD significantly increased upon treatment with 1,6-HD, with the distances in recovered cells returning to levels similar to those seen in untreated cells (Figs 2B and C and S2A and Table S2). In contrast, there was no significant effect on interprobe distances at the nearby control non-polycomb target region (Figs 2E and S2B and Table S2). Treatment with 2,5-HD had no significant effect at either locus. 1,6-HD also resulted in increased distances separating the ends of HoxB, with the cluster also returning to a more compact conformation in recovered cells (Fig S2C and Table S2).

The influence of 1,6-HD on chromatin compaction at HoxB and HoxD, but not at a nearby control locus, strongly implicates a specific effect of 1,6-HD on polycomb-marked chromatin. To investigate this further, we carried out the same analysis on *Ring1B$^{-/-}$* mESCs (Fig 2F). As we have shown previously (Eskeland et al, 2010;

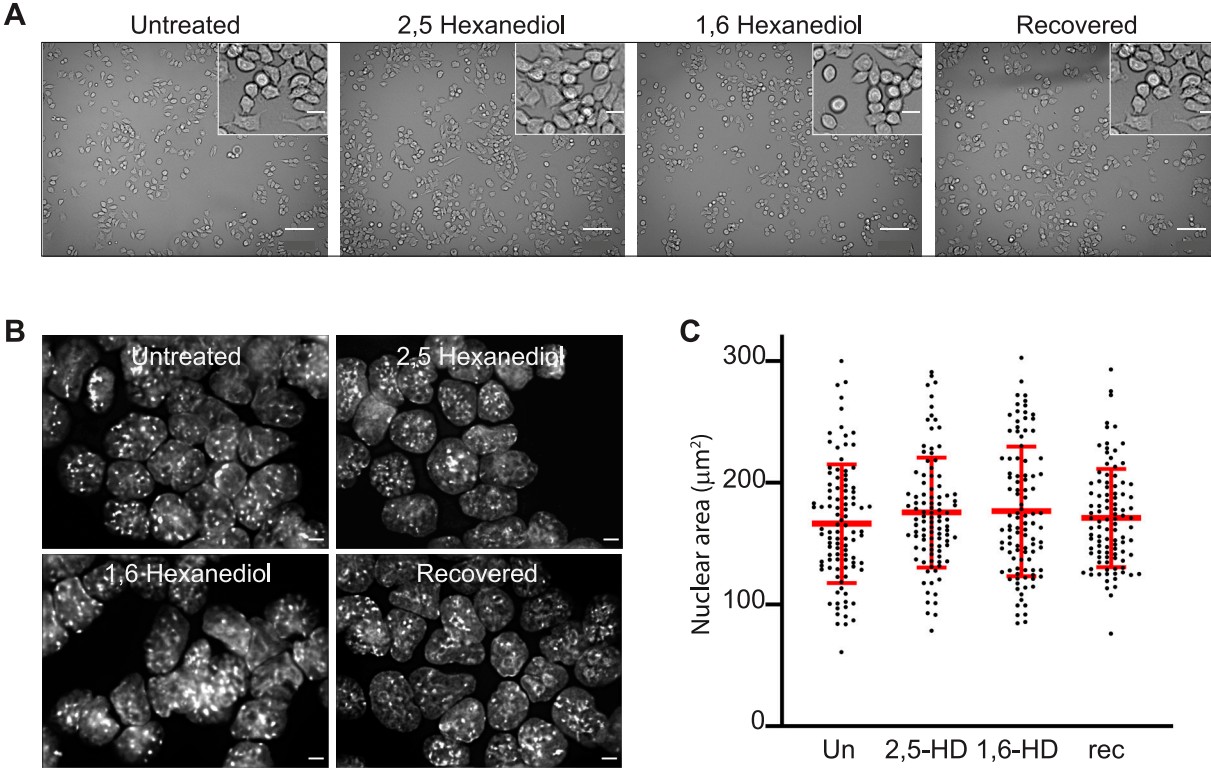

**Figure 1. Optimal hexanediol application conditions for mESCs are 2% concentration for 5 min.**
**(A)** Representative phase contrast images of untreated, 2,5-HD–treated, 1,6-HD–treated, and recovered mESCs. Scale bars: 100 μm. Inset scale bars: 20 μm.
**(B)** Representative images of DAPI-stained untreated, 2,5-HD–treated, 1,6-HD–treated, and recovered mESC nuclei. Scale bars: 5 μm. **(C)** Scatter plot showing the nuclear areas (μm²) of untreated, 2,5-HD–treated, 1,6-HD–treated, and recovered mESCs (black dots). Red bars show the means and standard deviations. Data sets were tested for differences using the unpaired t test with Welch's correction. All P-values were ≥0.1. Results from a biological replicate experiment are shown in Fig S1, and the data and statistical evaluation for these figures are in Table S1.

Boyle et al, 2020), the HoxD cluster is in a decompact conformation in *Ring1B*⁻/⁻ mESCs compared with wild-type (WT) cells (Fig 2G and Table S2). Treatment of *Ring1B*⁻/⁻ mESCs with 1,6-HD had no further effect on interprobe distances measured across the HoxD cluster (Fig 2G and Table S2). These results suggest that disruption of weak multivalent interactions causes a specific and reversible loss of chromatin compaction at Hox loci, that is, dependent on cPRC1.

### Reversible disruption of dispersed homeobox gene clusters by 1,6-HD

In addition to causing local chromatin compaction, PRC1 has also been implicated in bringing together more distal polycomb targets (Isono et al, 2013; Schoenfelder et al, 2015; Vieux-Rochas et al, 2015; Kundu et al, 2017; Boyle et al, 2020). Whereas the *Hox* genes are densely packed within the ~100 kb of each of the Hox loci, other clusters of vertebrate genes encoding homeodomain transcription factors are more dispersed. An example of the latter are the *Irx* gene clusters encoding the Iroquois transcription factors (Peters et al, 2000). *Irx3*, *Irx5*, and *Irx6* are spread across 900 kb of mouse chromosome 8, albeit within the same large (~2 Mb) TAD (Sobreira et al, 2021). In mESCs, all three genes are polycomb targets and form polycomb (RING1B)-dependent associations with each other in Hi-C (Fig 3A) (Boyle et al, 2020) and micro-C data, and these associations

are unaffected by the loss of either CTCF or cohesin (Fig S3A) (Hsieh et al, 2020, 2022). This association is reflected in the tight nuclear co-localisation of *Irx3*, *Irx5*, and *Irx6* as determined by DNA-FISH (Fig 3B). Two out of three of *Irx3*, *Irx5*, and *Irx6* are defined as clustered (<200 nm separation) at a quarter of measured alleles (Figs 3C and S3B and C and Table S3). Indeed, for a large proportion of alleles, all three loci cluster within 400 nm of each other (Fig 3D). All three loci were separated by >400 nm in <20% of alleles.

These proportions were not significantly different in 2,5-HD–treated cells. However, *Irx* clustering was significantly disrupted in 1,6-HD–treated mESCs cells (Figs 3C and D and S3B and C and Table S3) and was restored after 1,6-HD removal (recovered). *Irx* clustering was low (7–13%) in *Ring1B*⁻/⁻ mESCs and most alleles contained *Irx3*, *5*, and *6* in more dispersed configurations. There was no further dispersal of these three loci in 1,6-HD–treated *Ring1B*⁻/⁻ cells (Figs 3E and F and S3D and Table S3). Therefore, the effect of 1,6-HD can be attributed to polycomb function.

### Reversible disruption of long-range inter-TAD polycomb interactions by 1,6-HD

Interactions between distal polycomb targets can occur over large genomic distances that span beyond a single TAD (Boyle et al, 2020). As an example, *Shh* located within a 900 kb TAD is one of only three

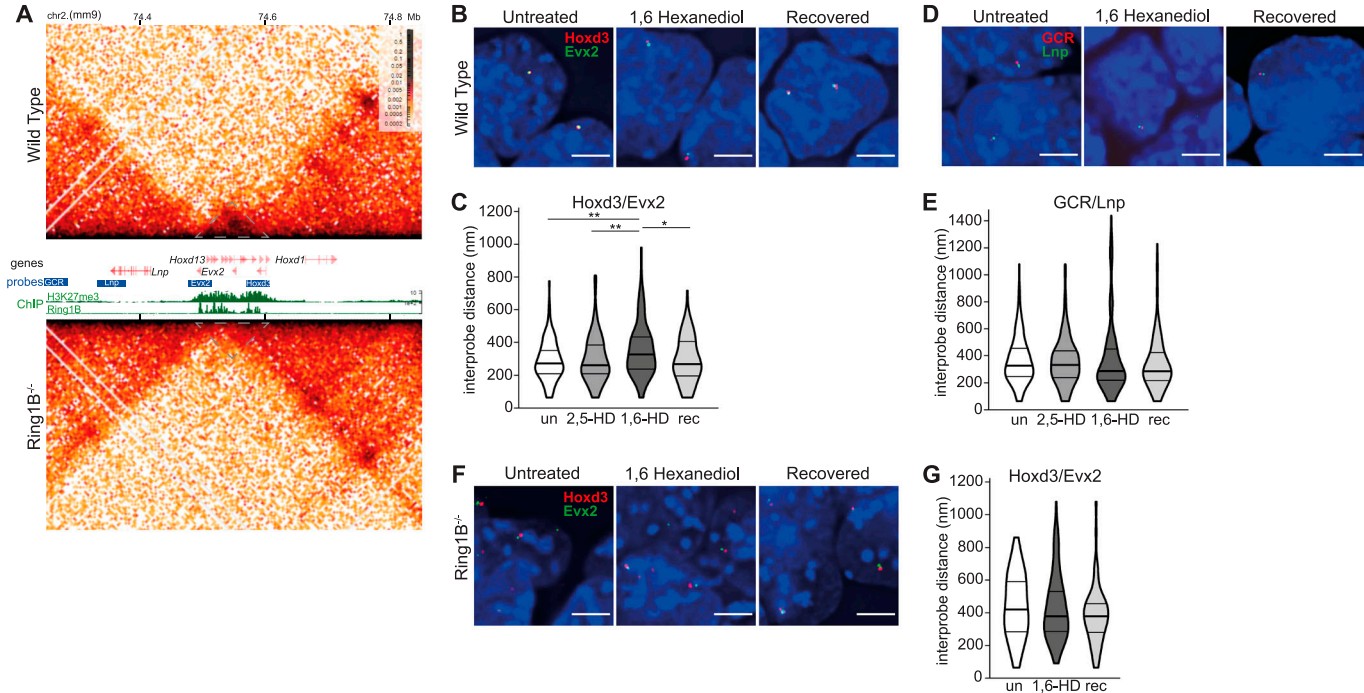

**Figure 2. Reversible chromatin decompaction at the HoxD cluster following 1,6-HD treatment.**
**(A)** Hi-C maps (5 kb resolution) of the HoxD cluster from wild-type (WT) (top: $Ring1B^{+/+}$) and (lower) $Ring1B^{-/-}$ mESCs. Genome co-ordinates on chromosome 2 (Mb) from the mm9 assembly of the mouse genome. Genes, fosmid probe–binding locations, H3K27me3, and RING1B ChIP-seq profiles are shown between the two Hi-C maps. Grey dotted triangles indicate the extent of the HoxD locus (data are from the study of Boyle et al [2020]). **(B, D)** Representative 3D-FISH images for polycomb (B) and non-polycomb (D) target loci at HoxD in untreated, 1,6-HD–treated, and recovered WT mESCs. Scale bars: 5 $\mu$m. **(C, E)** Violin plots show the interprobe distances (nm) for probe pairs used in B, D in untreated, 2,5-HD–treated, 1,6-HD–treated, and recovered WT cells. ∗ $P \le 0.05$ and > 0.01; ∗∗ $P \le 0.01$; Mann–Whitney test. Data for a biological replicate are shown in Fig S2, and data and statistical evaluation are summarised in Table S2. **(F, G)** As in B, C but using $Ring1B^{-/-}$ mESCs.

polycomb target genes across a 2 Mb region of chromosome 5 in mESCs, the other two genes—*En2* and *Mnx1*—being in the flanking TADs (Fig 4A). Hi-C (Fig 4A) and Micro-C (Fig S4A) (Hsieh et al, 2020, 2022) identify enriched interactions between *En2*, *Shh*, and *Mnx1* that are lost in Ring1B null cells (Fig 4A) (Boyle et al, 2020) but that are unaffected by the loss of CTCF or RAD21 (Fig S4A). Spatial proximity between all three polycomb target genes could also be detected, which was significantly reduced in Ring1B null cells, or when one of the three genes is expressed in vivo (Boyle et al, 2020). We performed 3D-FISH, using probes covering polycomb target and adjacent non-target loci (Boyle et al, 2020) (Fig 4A), on untreated, 1,6-HD– and 2,5-HD–treated, and recovered mESCs (Fig 4B). As was the case for the intra-TAD polycomb associations at the *Irx* locus, clustering of *Shh*, *En2*, and *Mnx1* was significantly reduced by 1,6-HD treatment, and restored after 1,6-HD removal (rec); whereas 2,5-HD treatment had no effect (Figs 4C and D and S4B and C and Table S4). Clustered and dispersed ratios remained unchanged at the adjacent non-polycomb target loci (*Cnpy1*, SFPE1, *Ube3c*) across the various treatments, with high proportions of dispersed alleles (Figs 4E and S4D and E and Table S5). Addition of 1,6-HD had no significant effect on the spatial positioning of the three polycomb target loci in $Ring1B^{-/-}$ cells, with no further dispersal of *En2*, *Shh*, and *Mnx1* (Figs 4F and S4F and Table S4).

There is a strong interaction between the TAD boundaries involving CTCF sites adjacent to *Shh* and ZRS limb enhancer (Williamson et al, 2016, 2019). To determine if the loss of spatial

proximity between *Shh*, *En2*, and *Mnx1* was influenced by a general perturbation of local chromatin organisation affecting non-polycomb associations we compared spatial distances between probes covering *Shh* and ZRS and their adjacent CTCF sites. No significant differences in spatial proximity between these two loci were identified in 2,5-HD, 1,6-HD, and recovered cells compared with untreated cells (Fig S4G and Table S6). Therefore, disruption of PRC1 condensates causes the specific dispersal of polycomb-dependent associations between three polycomb target genes located across three adjacent TADs over a 2 Mb distance.

## Hexanediol treatment reversibly perturbs chromosome-wide clustering of polycomb targets

Previously, we showed that cPRC1 has a constraining influence on chromosomal architecture through extreme long-range, chromosome-wide, clustering of polycomb target loci (Boyle et al, 2020). To determine whether cPRC1 associations at this length scale are also sensitive to disruption by 1,6-HD, we used two sets of oligo probe pools, one labelling polycomb target loci (PcG+), the other intervening non-polycomb target regions (PcG−) covering 28 discrete loci along 51 Mb of chromosome 2 (Fig 5A).

As in the absence of PRC1, the addition of 1,6-HD caused significantly reduced clustering of polycomb target loci but not for the intervening non-polycomb regions (Fig 5B and C and Table S7). 2,5-HD treatment caused a more subtle disruption of polycomb target

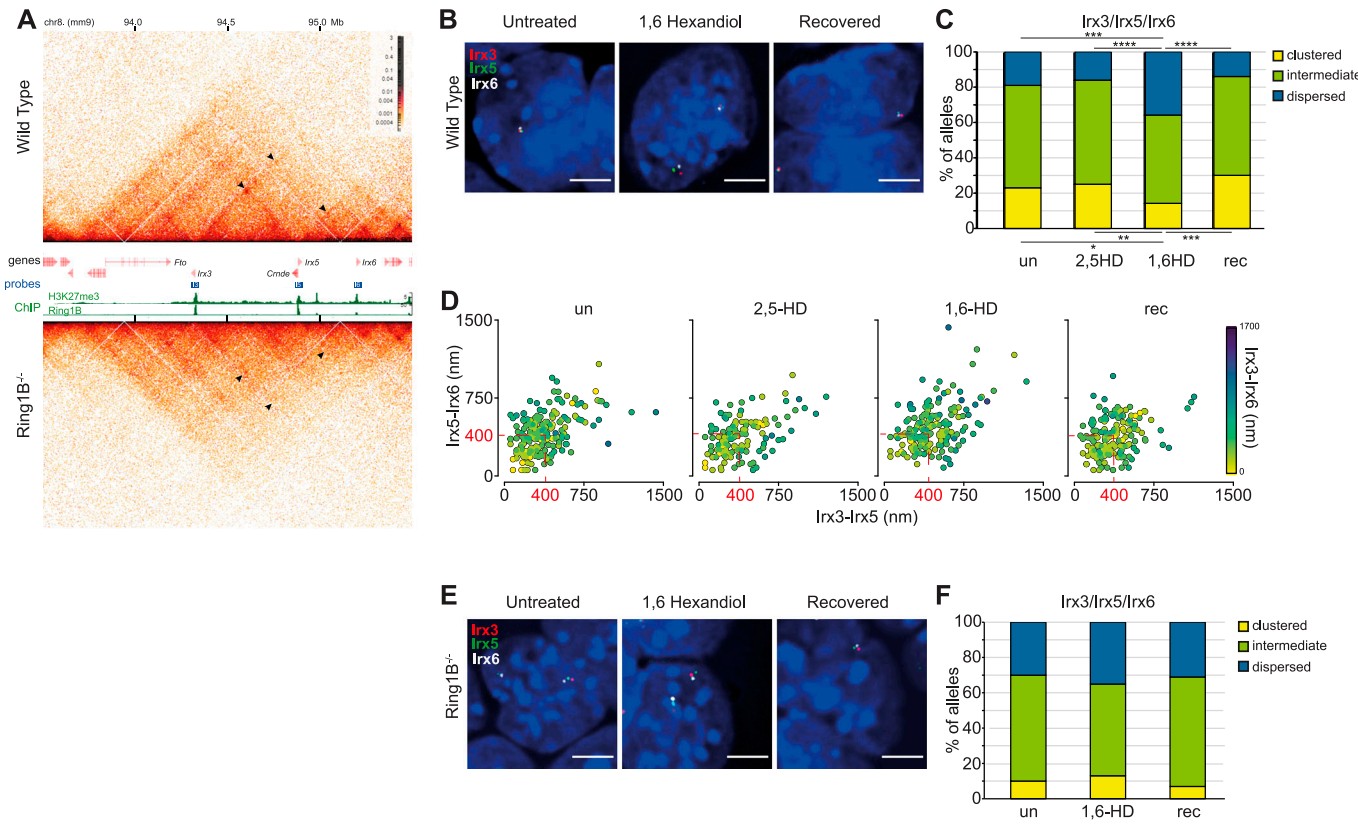

**Figure 3. Reversible disruption of *Irx* gene clustering by 1,6-HD.**
**(A)** Hi-C maps of the TAD on chromosome 8 (mm9 co-ordinates in Mb) containing *Irx3, 5,* and *6* from (upper) WT (*Ring1B⁺/⁺*) and (lower) *Ring1B⁻/⁻* mESCs at 5 kb resolution. Genes, fosmid probe–binding locations, H3K27me3, and RING1B ChIP-seq profiles are shown between the two Hi-C maps (data are from the study of Boyle et al [2020]). Black arrowheads indicate enriched interactions between *Irx3, 5,* and *6* in WT mESCs and the reduced interactions in *Ring1B⁻/⁻* cells. **(B)** Representative 3D-FISH images of *Irx3, 5,* and *6* loci in untreated, 1,6-HD–treated and recovered WT mESCs. Scale bar: 5 μm. **(C)** Bar plots providing categorical analysis of the spatial relationship of *Irx3, 5,* and 6 in untreated, 2,5-HD–treated, 1,6-HD–treated and recovered WT cells. Categories are as follows: clustered, at least two of the three loci <200 nm apart; intermediate, at least two of the three loci between 200–399 nm apart; dispersed, all three loci ≥400 nm apart. Differences in clustering and dispersal are identified using Fisher's exact test; ∗ P ≤ 0.05 and > 0.01, ∗∗ P ≤ 0.01, ∗∗∗ P ≤ 0.001, ∗∗∗∗ P ≤ 0.0001. **(D)** Scatter plots showing interprobe distances between each of the two probe pairs indicated on *x* and *y* axes with the separation between the third pair indicated by the color (in the color bar) in untreated, 2,5-HD–treated, 1,6-HD–treated, and recovered WT mESCs. Dashed red box indicates non-dispersed alleles with at least two sets of interprobe distances <400 nm. **(E, F)** As for (B, C), but for *Ring1B⁻/⁻* cells. Scatter plots for those data are in Fig S3, as are data for a biological replicate. All data and statistical evaluation are summarised in Table S3.

clustering but the addition of this alcohol also reduced clustering of the non-polycomb intervening loci, suggesting a chromosome-wide effect of this alcohol (Fig 5C and Table S7). Surprisingly, enhanced clustering of both PcG+ and PcG– probe pools was detected in recovered cells, compared with untreated cells (Fig 5B and C and Table S7). This suggests a persistent effect of 1,6-HD at the chromosome territory level.

## Perturbation of chromatin organisation by 1,6-HD is not simply the consequence of loss of PRC1 binding

The influence of PRC1 on multiple levels of chromatin organisation, from individual chromatin domains such as the Hox clusters to clustering of polycomb target loci over various length scales, has been demonstrated by the perturbation of these structures in *Ring1B⁻/⁻* cells (Boyle et al, 2020). To determine if the specific effects of 1,6-HD, and not 2,5-HD, on these same levels of 3D chromatin organisation could be attributed to the loss of PRC1 binding to chromatin following treatment, we performed calibrated ChIP-seq

(cChIP-seq [Hu et al, 2015]) for RING1B in mESCs treated with 1,6-HD, or 2,5-HD, and in 1,6-HD–recovered cells, noting however that this does not distinguish between RING1B that is in cPRC1 versus vPRC1 complexes.

Substantially reduced RING1B occupancy was detected genome-wide at target loci in cells treated with both 2,5-HD and 1,6-HD (Figs 6A and S5A). RING1B occupancy levels were restored in recovered cells. This was also the case when examining specific loci (Figs 6B and C and S5B and C). We saw no evidence for preferential retention of RING1B at specific sites in 1,6-HD– or 2,5-HD–treated cells.

Quantitative ChIP-PCR confirmed the reduction of RING1B occupancy after 1,6-HD and 2,5-HD treatment (particularly at the single polycomb target gene *En2*), and then its recovery, at *Hoxd10* and *En2* (Fig 6D). This was not accompanied by any significant differences in levels of the PRC2-catalyzed histone mark H3K27me3 at these same loci (Fig 6E). We conclude that the specific effects of 1,6-HD on chromatin compaction and long-range association between loci that are targets of cPRC1—effects not seen with 2,5-HD—cannot

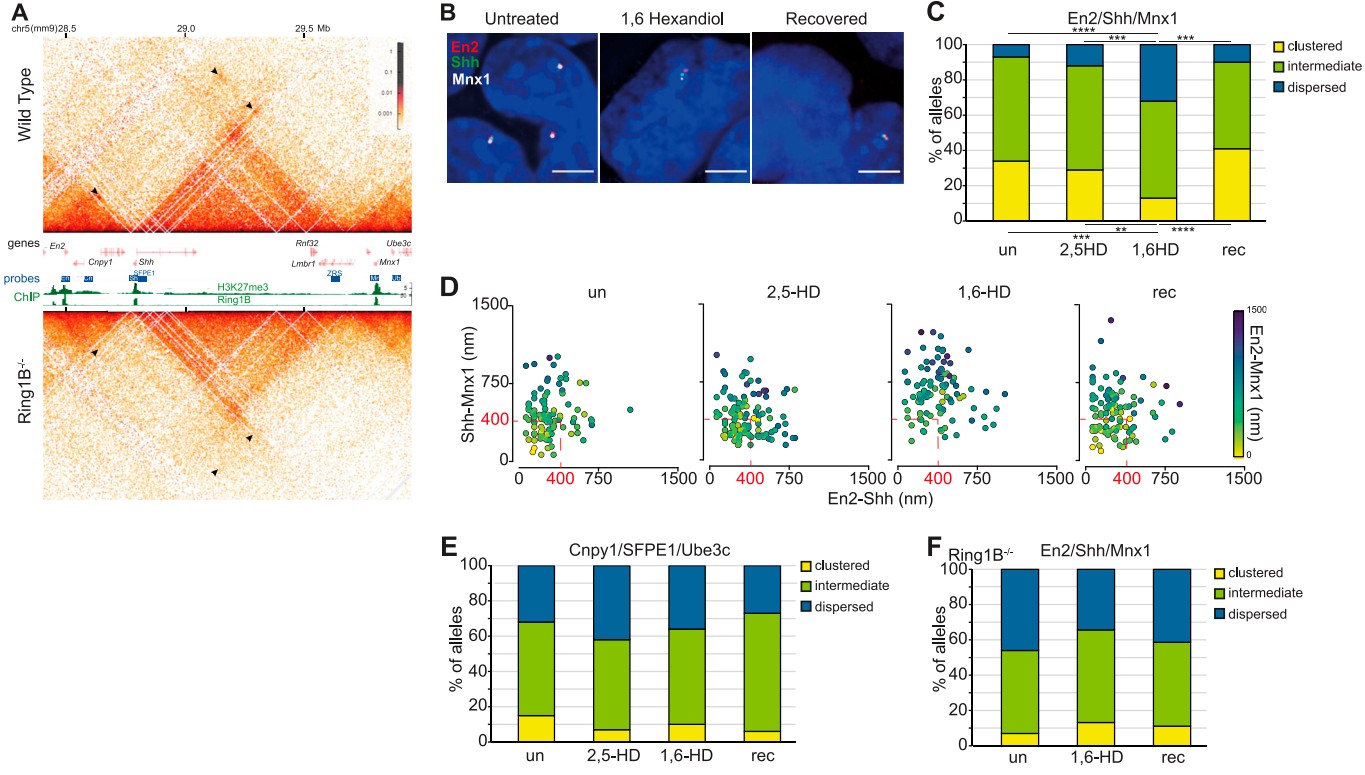

**Figure 4. Reversible disruption of inter-TAD polycomb clustering by 1,6-HD.**
**(A)** Hi-C maps of the three adjacent TADs on chromosome 5 (mm9) containing *En2*, *Shh*, and *Mnx1* polycomb targets from WT (*Ring1B*⁺/⁺) and *Ring1B*⁻/⁻ mESCs at 5 kb resolution. Genes, fosmid probe–binding locations, H3K27me3, and RING1B ChIP-seq profiles are shown between the two Hi-C maps (data from the study of Boyle et al [2020]). Black arrowheads indicate enriched interactions between the three polycomb target genes in WT cells and reduced interactions in *Ring1B*⁻/⁻ cells. **(B)** Representative images of 3D-FISH for *En2*, *Shh*, and *Mnx1* in untreated, 1,6-HD–treated, and recovered WT cells. **(C, E)** Bar plots providing categorical analysis of the spatial location of each of the three polycomb (C) and non-polycomb (E) probes shown in *A* relative to each other in untreated, 2,5-HD–treated, 1,6-HD–treated, and recovered WT cells. Categories are as in Fig 3. Differences in clustering and dispersal identified using Fisher's exact test; ** *P* ≤ 0.01, *** *P* ≤ 0.001, **** *P* ≤ 0.0001. **(D)** Scatter plots depicting the interprobe distances between each of the two fosmid probe pairs indicated on *x* and *y* axes with the separation between the third pair indicated by the color in the color bar for polycomb target loci in WT cells. Dashed red box indicates non-dispersed alleles with at least two sets of interprobe distances <400 nm. **(F)** As in C but for untreated, 1,6-HD–treated, and recovered *Ring1B*⁻/⁻ cells. Data for a biological replicate are in Fig S4. Data and statistical evaluation are summarised in Tables S4 and S5.

simply be attributed to reduced RING1B chromatin binding because that is affected by both alcohols.

## Discussion

Because PRC1 can mediate multivalent chromatin interactions that strongly influence local and long-range 3D genome organisation (Isono et al, 2013; Schoenfelder et al, 2015; Vieux-Rochas et al, 2015; Kundu et al, 2017; Boyle et al, 2020), and subunits of canonical PRC1 can undergo LLPS (Plys et al, 2019; Tatavosian et al, 2019; Seif et al, 2020; Eeften et al, 2021), it is plausible that the 3D nuclear organisation of polycomb target loci in mammalian cells involves the formation of biomolecular condensates through processes that could include, but that are not limited to, phase separation. In support of this hypothesis, here, we have shown by DNA-FISH in mESCs that PRC1-mediated compact chromatin domains, and the clustering of polycomb targets at length scales of many Mb, are reversibly perturbed by 1,6 hexanediol, but not by 2,5-HD. 1,6-HD has been reported to have very broad effects on nuclear transport,

enzyme function, chromatin motion, and to cause chromatin hypercondensation (Patel et al, 2007; Duster et al, 2010; Itoh et al, 2021). However, it is hard to attribute such non-specific effects to the very specific effects of 1,6-HD we detect at polycomb target loci and not at non-polycomb loci, and their dependence on functional PRC1 (*Ring1B*⁻/⁻). Indeed, impeded chromatin motion has also been reported following 2,5-HD treatment (Itoh et al, 2021), which we show does not affect polycomb chromatin compaction or local interactions. We conclude that these layers of polycomb-dependent 3D chromatin organisation are specifically sensitive to disruption by 1,6-HD. Because of our focus on polycomb domains, we cannot comment on the reported effects of 1,6-HD on aspects of 3D genome organisation assayed by Hi-C, such as compartments, TADs, and enhancer–promoter interactions (Liu et al, 2021; Ulianov et al, 2021).

A general increase in short-range contact frequencies assayed by Hi-C has been reported after short duration 1,6-HD treatment (Liu et al, 2021). However, our imaging data detect very specific disruption of local compaction (e.g., *Hox* loci) and focal interactions between polycomb target loci that are within a few 100 kb of each

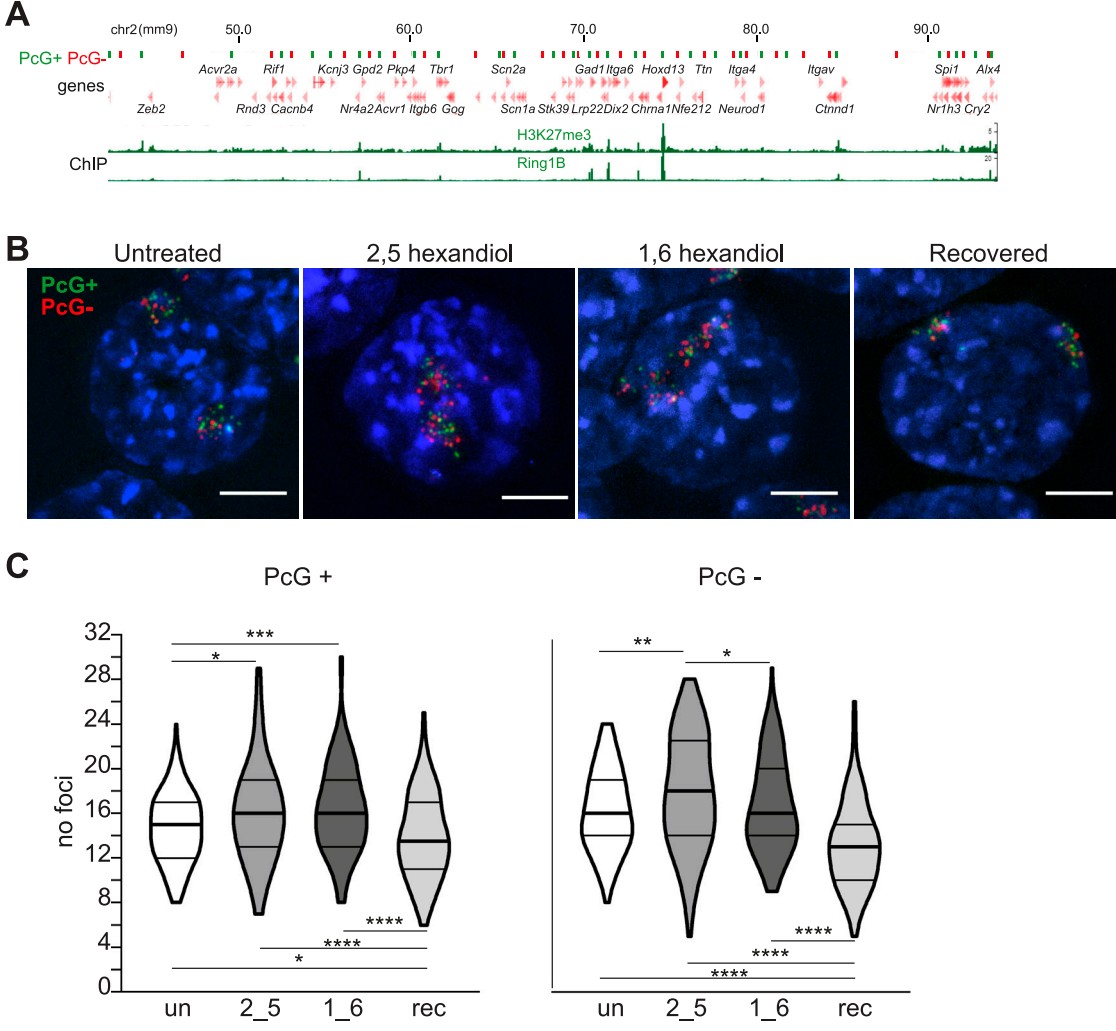

**Figure 5. Reversible loss of PRC1-dependent clustering of polycomb targets across chromosome 2 following hexanediol treatment.**
**(A)** Ideogram of chromosome 2 indicating the location of the oligonucleotide probes used in B, C and zoomed in browser tracks of RING1B and H3K27me3 ChIP-seq from WT mESCs (Boyle et al, 2020). Polycomb-positive (PcG+) and polycomb-negative (PcG−) probe locations are represented as green and red bars, respectively. Genomic co-ordinates (Mb) are for the mm9 genome assembly. **(B)** Representative 3D-FISH images of untreated, 2,5-HD–treated, 1,6-HD–treated, and recovered cells hybridized with the chromosome 2 PcG+ (green; 6FAM) and PcG− (red; ATT0594) probe pools. Scale bar: 5 µm. **(C)** Violin plots depicting the number of discrete foci in untreated, 2,5-HD–treated, 1,6-HD–treated, and recovered cells for PcG+ and PcG− probe pools. ∗ $P ≤ 0.05$ and $> 0.01$; ∗∗ $P ≤ 0.01$; ∗∗∗ $P ≤ 0.001$, ∗∗∗∗ $P ≤ 0.0001$; Mann–Whitney test. Data are summarised in Table S7.

other (*Irx3*, *5*, and *6* in Fig 3/*Shh* and *En2* in Fig 4) by short-term 1,6-HD treatment.

Using calibrated ChIP, we show that 1,6-HD significantly reduces, but does not completely abolish, the occupancy of the PRC1 subunit RING1B at polycomb target loci genome-wide that can be captured by formaldehyde cross-linking. However, reduced PRC1 occupancy as assayed by ChIP also occurs in cells treated with 2,5-HD, a similar alcohol that does not have the same disruptive effects on chromatin compaction and polycomb target clustering as 1,6-HD. Although we cannot formally exclude that both 1,6-HD and 2,5-HD alter the kinetics of RING1B chromatin binding, and hence the ability of formaldehyde to capture the chromatin bound fraction (Schmiedeberg et al, 2009), or disrupt the structure of the PRC1 complex itself, we attribute the loss of 1,6-HD–sensitive interactions

between PRC1 complexes, rather than reduced chromatin binding, as primarily responsible for the perturbations of polycomb domain compaction and focal interactions we have identified. Unlike conclusions drawn from ectopically expressed proteins (Eeften et al, 2021), our results suggest that chromatin compaction by PRC1 is dependent on condensate formation. The continued presence of H3K27me3 at polycomb loci after 1,6-HD treatment also excludes the possibility that a more generalised disruption of histone occupancy is responsible for the changes in 3D chromatin conformation.

We did find that extremely long-range (chromosome-wide) interactions between polycomb sites were impacted by 2,5-HD treatment, although to a lesser extent than for 1,6-HD (Fig 5). Because 2,5-HD also seemed to decrease binding of PRC1 genome-wide, this result suggests that interactions of polycomb sites over very large

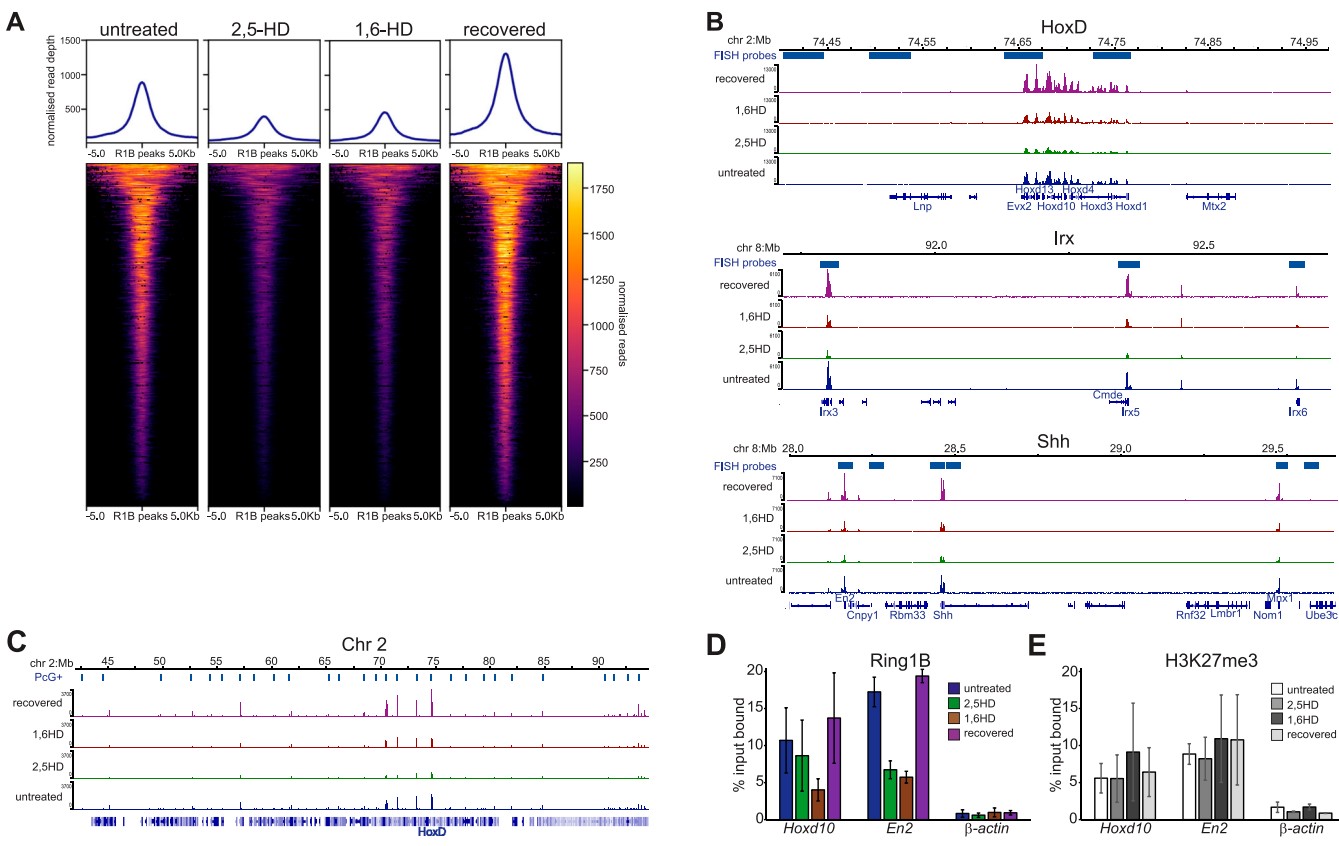

**Figure 6. Reversible reduction of PRC1 binding to polycomb target loci following hexanediol treatment.**
**(A)** Heatmap representation (bottom panel) and average metaplots (top panel) of RING1B calibrated ChIP-seq signal distribution at merged RING1B peaks in untreated, 2,5-HD–treated, 1,6-HD–treated, and recovered WT mESCs. **(B)** IGV (Robinson et al, 2011) track views of RING1B cChIP-seq data from untreated, 2,5-HD–treated, 1,6-HD–treated, and recovered mESCs at the HoxD (*top*), *Irx3, 5, 6* (*middle*), *En2, Shh*, and *Mnx1* (*bottom*) genomic regions. **(C)** As for B but for the 50 Mb region of chromosome 2 tiled with oligonucleotide probes. Biological replicate for these data are in Fig S5. Genomic co-ordinates (Mb) are for the mm10 genome assembly. **(D, E)** qPCR analysis of ChIP for RING1B (D) and H3K27me3 (E) at *Hoxd10, En2*, and *β*-actin in untreated, 2,5-HD-treated, 1,6-HD–treated, and recovered mESCs. Enrichment is shown as mean percent input bound ± s.e.m. over two (*β*-actin) to four biological replicates.

length scales that are probably initiated through stochastic interactions in nuclear space, may rely, at least in part, on there being sufficiently high levels of PRC1 bound at target sites. The effect of 2,5-HD treatment on 3D genome organisation over very large length scales may be due to the reported effects of this alcohol on global chromatin motion (Itoh et al, 2021).

To preserve mESC morphology and cell viability, here we have only been able to induce very transient (5 min) disruption of polycomb-dependent 3D genome folding using 1,6-HD. Therefore, we are not able to determine the contribution that polycomb-mediated condensates may play in epigenetic mechanisms. Investigation of this interesting question awaits the development of less blunt tools for perturbing condensates in vivo.

# Materials and Methods

### Cell culture and alcohol treatments

Feeder-free mESCs E14tg2A (129/Ola; *Ring1B*$^{+/+}$) and the derivative line (*Ring1B*$^{-/-}$) (Illingworth et al, 2015) were cultured at

37°C on 0.1% gelatin-coated (G1890; Sigma-Aldrich) Corning flasks in GMEM BHK-21 (21710-025; Gibco) supplemented with 10% FCS(F-7524; Sigma-Aldrich), LIF, nonessential amino acids (11140-035; Gibco), sodium pyruvate (11360-039; Gibco), 50 $\mu$M2 $\beta$-mercaptoethanol (31350-010; Gibco), and L-glutamine. For passaging, 60–90% confluent flasks were washed with PBS, incubated for 2–3 min (mins) at room temperature in 0.05% (vol/vol) trypsin (25300-054; Gibco), and tapped to release. Trypsin was inactivated by adding 9 vol/vol of ESC medium, and this mixture was repeatedly pipetted to obtain a single-cell suspension. ESCs were centrifuged, resuspended in ESC medium, and replated on to gelatin-coated flasks at a density of ~4 × 10$^4$ cells/cm$^2$. For optimization of hexanediol experiments, ~4 × 10$^4$ cells/cm$^2$ or ~1 x 10$^6$ cells were plated in standard medium, into flasks (for ~48 h) or on gelatin-coated SuperFrost plus microscopic glass slides (~24 h), respectively, before replacement with standard medium supplemented with 1,6 hexanediol at 2%, 5%, and 10% concentrations (wt/vol) for 5, 10, or 15 min, respectively. For subsequent experiments cells growing in flasks and on glass slide–plated cells were incubated in 2% 1,6 hexanediol (wt/vol) and 2,5 hexanediol (vol/vol) for 5 min before harvesting or fixation. Medium containing 1,6 hexanediol was replaced with

standard medium and cells were cultured for at least 1 h (hr) before the harvesting or fixing of recovered cells.

All centrifugation steps with live cells were performed at 330*g* for 4 min at room temperature. All ESC lines used in this study were routinely tested for mycoplasma.

### 3D DNA-FISH

#### *Fixation*

Mouse ESCs grown on slides were fixed in 4% PFA for 10 min, permeabilized in PBS/0.5% Triton X-100, dried, and then stored at −80°C before hybridization. Slides were incubated in 100 $\mu$g/mL RNase A in 2× SSC for 1 h at 37°C, washed briefly in 2× SSC, passed through an alcohol series, and air-dried. Slides were incubated for 5 min at 70°C, denatured in 70% formamide/2× SSC (pH 7.5) for 40 min at 80°C, cooled in 70% ethanol for 2 min on ice, and dehydrated by immersion in 90% ethanol for 2 min and 100% ethanol for 2 min before air drying.

#### *Hybridization and detection*

For each slide hybridized with oligo probes, 800 nanograms of each fluorescently labelled oligonucleotide probe pool (2 $\mu$l; MyTags, Tables S8 and S9) (Boyle et al, 2020) were added to 26 $\mu$l of hybridization mix (50% formamide, 2× SSC, 1% Tween 20, 10% dextran sulfate), denatured for 5 min at 70°C, and then snap-chilled on ice. For each slide hybridized with fosmid probes, 160–240 ng of biotin-, digoxigenin-, and red-dUTP labelled (Alexa fluor 594-dUTP [Invitrogen]) fosmid probes (Table S10), with 16–24 $\mu$g of mouse Cot1 DNA, and 10 $\mu$g of sonicated salmon sperm DNA were dried in a Spin-Vac and then reconstituted in 30 $\mu$l of hybridization mix. Probes were then denatured for 5 min at 80°C, reannealed for 15 min at 37°C.

Fosmid and oligonucleotide probes were hybridized to slides under a sealed coverslip overnight at 37°C. Slides were washed the next day four times for 3 min in 2× SSC at 45°C and four times for 3 min in 0.1× SSC at 60°C. Slides hybridized with directly labelled oligonucleotide probes were immediately stained with 4,6-diaminidino-2- phenylidole (DAPI) at 50 ng/mL, mounted in VECTA-SHIELD (Vector Laboratories), and sealed with nail varnish. Slides hybridized with biotin- and dig-labelled fosmid probes were blocked in blocking buffer (4 x SSC, 5% Marvel) for 5 min. The following antibody dilutions were made: fluorescein anti-dig FAB fragments (cat. no. 11207741910; Roche) 1:20, fluorescein anti-sheep 1:100 (cat. no. FI-6000; Vector)/streptavidin Cy5 1:10 (cat. no. PA45001, lot 17037668; Amersham), biotinylated anti-avidin (cat. no. BA-0300, lot ZF-0415; Vector) 1: 100, and streptavidin Cy5 1:10. Slides were incubated with antibody in a humidified chamber at 37°C for 30–60 min in the following order with 4X SSC/0.1% Tween 20 washes in between: fluorescein anti-dig, fluorescein anti-sheep/streptavidin Cy5, biotinylated anti-avidin, streptavidin Cy5 before staining and mounting with DAPI and VECTASHIELD, respectively.

#### *Imaging and image analysis*

Epifluorescent images were acquired using a Photometrics Cool-SNAP HQ2 CCD camera and a Zeiss AxioImager A1 fluorescence microscope with a plan apochromat 100× 1.4 NA objective, a Lumen 200-W metal halide light source (Prior Scientific Instruments) and Chroma 89014ET single-excitation and emission filters (three-color FISH) or Chroma 89000ET single-excitation and emission filters (four-color FISH) (Chroma Technology Corp.) with the excitation and emission filters installed in Prior motorized filter wheels. A piezoelectrically driven objective mount (PIFOC model P-721; Physik Instrumente GmbH & Co.) was used to control movement in the z-dimension. Hardware control, image capture, and analysis were performed using Volocity (Perkinelmer, Inc.) or Nis elements (Nikon). Images were deconvolved using a calculated point spread function with the constrained iterative algorithm of Volocity. Image analysis was carried out using the quantitation module.

### Hi-C and Micro-C data

Hi-C data from *Ring1B*$^{+/+}$ and *Ring1B*$^{-/-}$ mESCs were taken from the study of Boyle et al (2020). Micro-C WT mESC data were taken from the study of Hsieh et al (2020) and Micro-C data from CTCF-AID and RAD21-AID IAA-treated ES cells were taken from the study of Hsieh et al (2022). Both Hi-C and Micro-C data sets were visualised at 5 kb resolution using HiGlass (Kerpedjiev et al, 2018).

### cChIP-seq

cChIP-seq was carried out as previously described (Boyle et al, 2020). Briefly, trypsinized untreated, 2,5-HD, 1,6-HD, and recovered mESCs (~2 × 10$^7$) were washed twice in PBS. After 10 min of formaldehyde (cat. #344198, final concentration 1%; Calbiochem) fixation, stopped by 5 min incubation with 125 mM of glycine at room temperature, cells were washed in PBS, pelleted, and stored at −80°C. Pellets were then thawed and combined with 1 × 10$^6$ formaldehyde-fixed human MCF-7 cells (for downstream calibration of ChIP-seq data). Following lysis, cells were sonicated using a cooled Bioruptor (50 cycles, 1 min cycles of 30 s on/30 s off on "high" setting at 4°C). The sonicated extract was pre-cleared by centrifugation at 16,000*g* for 10 min at 4°C. The supernatant was transferred to a fresh tube and supplemented with BSA to a final concentration of 25 mg/mL. A sample of the chromatin was retained as an input reference.

Antibodies were precoupled to a 1/1 mixture of protein A (10001D; Life Technologies) and protein G (1003D; Life Technologies) Dynabeads at a ratio of 1 mg antibody per 30 mL of Dynabead suspension by rotation for 2 h at 4°C. Cell equivalents (~6.5 × 10$^6$) of lysate were added to 7.5 $\mu$g of anti-RING1B (D22F2; Cell Signaling), 5 $\mu$g of anti-H3K27me3 (C36B11; Cell Signaling), or 5 $\mu$g rabbit IgG (Cell Signaling), respectively, and incubated overnight on a rotating wheel at 4°C. Following incubation, bead-associated immune complexes were washed sequentially with ChIP dilution buffer, wash buffer A, and wash buffer B, each for 10 min at 4°C on a rotating wheel, followed by two washes in TE buffer at room temperature. Chromatin was released by incubating the beads in 100 $\mu$l of elution buffer (0.1 M NaHCO3, 1% SDS) for 15 min at 37°C, followed by the addition of 20 $\mu$g of RNase A and 6 $\mu$l of 2 M Tris (pH 6.8) and incubation for 1 h at 37°C and finally by the addition of 20 $\mu$g proteinase K and incubation overnight at 65°C to degrade proteins and reverse the cross-links. Dynabeads were removed using a magnetic rack and the chromatin purified using PCR purification columns (QIAGEN) according to the manufacturer's instructions.

**Life Science Alliance**

Libraries (RING1B ChIP and corresponding input samples only) were constructed using the NEBNext Ultra II DNA library preparation kit for Illumina according to the manufacturer's instructions (E7645S; NEB). To determine the number of PCR cycles required for amplification, one aliquot of library preparation from each sample were supplemented with EvaGreen so that amplification could be monitored by quantitative PCR on a Bio-Rad C1000 Touch Thermal Cycler. To allow for sample multiplexing, PCRs were performed using index primers (NEBNext Multiplex Oligos for Illumina, set 1, E7335 and amplified to linear phase). Size selection purifications following the ligation and amplification PCR steps were performed with 0.9× reaction volumes of Agencourt AMPure XP beads (A63880; Beckman Coulter). Purified libraries were combined as an 8-sample equimolar pool containing a combination of indexes from 1–12 and sequenced on an Illumina NextSeq on an Illumina NextSeq 2000 on a P2 flow cell (paired-end 75-bp reads).

### ChIP-seq analysis

cChIP-seq data analysis was performed using the Nextflow (version 22.04.0) pipeline chip_quant_analysis.nf (https://gitbio.ens-lyon.fr/LBMC/Bernard/quantitative-nucleosome-analysis) with the reference genome *Mus musculus* (GRCm38) and for internal calibration the *Homo sapiens* (GRCh38) genome.

Briefly, FASTQ files were processed with fastp (0.20.1, options --qualified_quality_phred 20 --disable_length_filtering--detect_adapter_for_pe). The processed FASTQ files were mapped to a concatenated reference and calibration genome with Bowtie2 (2.3.4.1, option --very-sensitive). Bam files were split into reads mapping exclusively to either the reference genome or the calibration genome, using samtools (1.11) and awk. Mapped reads were deduplicated using picard (2.18.11, options MarkDuplicates VALIDATION_STRINGENCY=LENIENT REMOVE_DUPLICATES=true). The deduplicated bam files were converted to bigwig and bedgraph formats using deepTools (3.5.1). Normalization and generation of ratio coverage files were generated using software and scripts contained in the docker containers lbmc/chip_quant_r:0.0.6 and biocontainers/danpos:v2.2.2_cv3 (Colin et al, 2022 *Preprint*).

Heatmaps were generated using the deepTools (Ramírez et al, 2014) function computeMatrix with settings "reference-point -a 5000 -b 5000" with spike-in normalised bigWig files and RING1B peaks from the study of Illingworth et al (2015) merged within 5 kb of each other. Plots were made with the deepTools function plotHeatmap.

### ChIP-qPCR analysis

Dilutions (fivefold) of sonicated input from each mESC sample (20% of each IP quantity) were used to create standard curves for the immunoprecipitated H3K27me3 and RING1B chromatin (two to four biological replicates). For each replicate, PCRs were performed in triplicate using LightCycler 480 SYBR Green I detection kit on a Bio-Rad C1000 Touch thermal cycler (using primers described in Table S11). Thermal cycler programme: 15 min Hotstart; 44 PCR cycles (95°C for 15 s, 55°C for 30 s, 72°C for 30 s).

### Statistical analysis

DNA-FISH interprobe distance data sets were compared using the two-tailed Mann–Whitney $U$ test, a nonparametric test that compares two unpaired groups. Differences in DNA-FISH data sets comparing categorical distributions were measured using Fisher's exact test. Nuclear size comparisons were measured using Unpaired $t$ test with Welch's correction, a parametric test that compares two unpaired groups. All statistical analyses were performed using GraphPad Prism 9.4.1 software (Mann–Whitney, $t$ test) or online GraphPad 2 x 2 contingency table www.graphpad.com/quickcalcs/contingency1/(Fisher's).

## Data Availability

All sequencing data have been submitted to the GEO repository under the accession number GSE224930.

## Supplementary Information

## Acknowledgements

The authors thank the staff of the Institute of Genetics and Cancer Advanced Imaging Resource for their assistance with imaging and the Wellcome Trust Clinical Research Facility (Edinburgh Clinical Research Facility) for sequencing. This work has made use of the resources provided by the Edinburgh Compute and Data Facility (ECDF) (http://www.ecdf.ed.ac.uk/). Work in the group of WA Bickmore is supported by MRC University Unit grants MC_UU_00007/2 and MC_UU_00035/7. ET Friman is supported by a fellowship from the Swiss National Science Foundation (P500PB_206805). For the purpose of open access, the author has applied a Creative Commons Attribution (CC BY) licence to any author-accepted manuscript version arising from this submission.

### Author Contributions

I Williamson: formal analysis, investigation, methodology, and writing—original draft, review, and editing.
S Boyle: investigation and visualization.
GR Grimes and ET Friman: formal analysis.
WA Bickmore: conceptualization, supervision, funding acquisition, project administration, and writing—original draft, review, and editing.

### Conflict of Interest Statement

The authors declare that they have no conflict of interest.

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
