## [Reviewer comments · Life Science Alliance]

Dispersal of PRC1 condensates disrupts polycomb chromatin domains and loops

Iain Williamson, Shelagh Boyle, Graeme Grimes, Elias Friman, and Wendy Bickmore
DOI: <https://doi.org/10.26508/lsa.202302101>

Corresponding author(s): Wendy Bickmore, MRC Human Genetics Unit

Review Timeline:	Submission Date:	2023-04-17
	Editorial Decision:	2023-04-18
	Revision Received:	2023-06-27
	Editorial Decision:	2023-07-04
	Revision Received:	2023-07-10
	Accepted:	2023-07-12

Transaction Report:

Please note that the manuscript was previously reviewed at another journal and the reports were taken into account in the decision-making process at *Life Science Alliance*. Since the original reviews are not subject to Life Science Alliance's transparent review process policy, the reports and author response cannot be published.

April 18, 2023

Re: Life Science Alliance manuscript #LSA-2023-02101-T

Wendy A Bickmore
MRC Human Genetics Unit

Dear Dr. Bickmore,

Thank you for submitting your manuscript entitled "Dispersal of PRC1 condensates disrupts polycomb chromatin domains and loops" to Life Science Alliance. We invite you to submit a revised manuscript addressing the following Reviewer points.

Thank you for this interesting contribution to Life Science Alliance. We are looking forward to receiving your revised manuscript.

Sincerely,

B. MANUSCRIPT ORGANIZATION AND FORMATTING:

July 4, 2023

RE: Life Science Alliance Manuscript #LSA-2023-02101-TR

Wendy Bickmore
MRC IGMM
Unknown

Dear Dr. Bickmore,

Thank you for submitting your revised manuscript entitled "Dispersal of PRC1 condensates disrupts polycomb chromatin domains and loops". We would be happy to publish your paper in Life Science Alliance pending final revisions necessary to meet our formatting guidelines.

- please add your main, supplementary figure, and table legends to the main manuscript text after the references section
- please add ORCID ID for the corresponding author--you should have received instructions on how to do so

A. FINAL FILES:

B. MANUSCRIPT ORGANIZATION AND FORMATTING:

Sincerely,

July 12, 2023

RE: Life Science Alliance Manuscript #LSA-2023-02101-TRR

Wendy Bickmore
MRC Human Genetics Unit
United Kingdom

Dear Dr. Bickmore,

Thank you for submitting your Research Article entitled "Dispersal of PRC1 condensates disrupts polycomb chromatin domains and loops". It is a pleasure to let you know that your manuscript is now accepted for publication in Life Science Alliance. Congratulations on this interesting work.

DISTRIBUTION OF MATERIALS:

Again, congratulations on a very nice paper. I hope you found the review process to be constructive and are pleased with how the manuscript was handled editorially. We look forward to future exciting submissions from your lab.

Sincerely,
